# Non-Parametric Statistical Approaches for Leaf Area Index Estimation from Sentinel-2 Data: A Multi-Crop Assessment

**Margherita De Peppo** [1,*], **Andrea Taramelli** [2,3], **Mirco Boschetti** [4], **Alberto Mantino** [1], **Iride Volpi** [1], **Federico Filipponi** [2], **Antonella Tornato** [2], **Emiliana Valentini** [2,5] **and Giorgio Ragaglini** [6]

1  Sant' Anna School of Advanced Studies, Institute of Life Sciences, Piazza Martiri della Libertà 33, 56127 Pisa, Italy; a.mantino@santannapisa.it (A.M.); i.volpi@santannapisa.it (I.V.)
2  Institute for Environmental Protection and Research (ISPRA), via Vitaliano Brancati 48, 00144 Rome, Italy; andrea.taramelli@iusspavia.it (A.T.); federico.filipponi@isprambiente.it (F.F.); antonella.tornato@isprambiente.it (A.T.); emiliana.valentini@cnr.it (E.V.)
3  Institute for Advanced Study of Pavia (IUSS), Palazzo del Broletto, Piazza della Vittoria 15, 27100 Pavia, Italy
4  Institute for Electromagnetic Sensing of the Environment, Italian National Research Council, Via Bassini 15, 20133 Milan, Italy; boschetti.m@irea.cnr.it
5  Institute of Polar Sciences, Italian National Research Council, via Salaria km 29,300, 00015 Rome, Italy
6  Dipartimento di Scienze Agrarie e Ambientali-Produzione, Territorio, Agroenergia, Università degli Studi di Milano, Via Celoria 2, 20133 Milano, Italy; giorgio.ragaglini@unimi.it
*  Correspondence: m.depeppo@santannapisa.it

**Abstract:** The leaf area index (LAI) is a key biophysical variable for agroecosystem monitoring, as well as a relevant state variable in crop modelling. For this reason, temporal and spatial determination of LAI are required to improve the understanding of several land surface processes related to vegetation dynamics and crop growth. Despite the large number of retrieved LAI products and the efforts to develop new and updated algorithms for LAI estimation, the available products are not yet capable of capturing site-specific variability, as requested in many agricultural applications. The objective of this study was to evaluate the potential of non-parametric approaches for multi-temporal LAI retrieval by Sentinel-2 multispectral data, in comparison with a VI-based parametric approach. For this purpose, we built a large database combining a multispectral satellite data set and ground LAI measurements collected over two growing seasons (2018 and 2019), including three crops (i.e., winter wheat, maize, and alfalfa) characterized by different growing cycles and canopy structures, and considering different agronomic conditions (i.e., at three farms in three different sites). The accuracy of parametric and non-parametric methods for LAI estimation was assessed by cross-validation (CV) at both the pixel and field levels over mixed-crop (MC) and crop-specific (CS) data sets. Overall, the non-parametric approach showed a higher accuracy of prediction at pixel level than parametric methods, and it was also observed that Gaussian Process Regression (GPR) did not provide any significant difference (*p*-value > 0.05) between the predicted values of LAI in the MC and CS data sets, regardless of the crop. Indeed, GPR at the field level showed a cross-validated coefficient of determination ($R^2_{CV}$) higher than 0.80 for all three crops.

**Keywords:** LAI; Sentinel-2; wheat; maize; alfalfa; parametric; non-parametric; GPR

## 1. Introduction

With the global increase in food demand, obtaining timely information regarding crop growth and retrieving detailed data of crop health have become essential aspects for developing strategic food policies and ensuring sustainable agroecosystem management. Reliable monitoring of crop yields at the regional scale can support policy makers in quantifying food supply (GEOGLAM initiative, https://www.earthobservations.org/geoglam.php, accessed on 20 April 2021), while mapping crop conditions at a field scale can assist farmers in agroecosystem management [1]. The leaf area index (LAI), defined as the total one-sided

area of leaf tissue per unit ground, is a biophysical indicator used to represent the dimension of the crop canopy and its variation over time [2]. Indeed, LAI measurements have been widely adopted for crop monitoring, as well as for modelling applications [3–5], being a key state variable associated with processes including light interception and soil–crop water balance [6–8]. Moreover, at the landscape level, LAI maps can provide information on cropping system status, according to crop rotation, soil coverage, crop phenological development, and their response to management and anomalies caused by extreme events in time (across seasons) and space (among and within fields) [4,9].

Remote sensing (RS) provides an effective way to retrieve LAI values at different spatial and temporal scales, as has been successfully demonstrated in different contexts [10–12]. According to Verrelst et al. [13], the approaches for biophysical parameter retrieval (e.g., LAI) can be classified in four main categories: (i) parametric regression methods, which assume an explicit relationship between spectral data (e.g., Vegetation Indices) and biophysical data (e.g., LAI); (ii) non-parametric regression methods, which do not require an explicit relationship and data distribution; (iii) physical-based methods, using radiative transfer models (RTMs) to simulate the interaction between spectral radiation and vegetation biophysical and biochemical parameters; and (iv) hybrid retrieval methods, combining non-parametric and physical-based approaches.

The first category relies on regression analyses (ground-LAI vs. VI) and their easy implementation for operational vegetation cover monitoring applications [13]. However, the drawbacks of VI-based methods are related to the implicit assumption that the reflectance variability depends only (or mainly) on the LAI [14]. Instead, canopy reflectance is strongly affected by several factors, such as the aboveground biomass, chlorophyll content, canopy architecture, and soil background, which vary in space and time, according to the crop phenology and seasonal conditions [15,16]. Therefore, parametric approaches are often crop- and site-specific, due to their dependence on the regression data set, thus making them inadequate for the general purpose of retrieving LAI values from a diversified landscape mosaic [17]. In this context, several VI formulations, according to different band compositions, have been developed, in order to cope with these limitations and to be better suited to mixed-crop scenarios [18–21]. For example, previous research has shown the potential of VIs based on red-edge (RE) and short-wave infrared regions (SWIR), in terms of being less sensitive to specific crop types than traditional Red/NIR-based indices (e.g., NDVI) [19,22–24]. Nevertheless, further investigations into the accuracy of parametric methods based on RE and SWIR vegetation indices are needed, in order to assess their accuracy for satellite remote sensing.

The second category refers to machine learning regression algorithms (MLRAs). MLRAs have gained widespread popularity, as they address the limitations of VI-based methods [24]. For this reason, different studies interested in LAI estimation have compared the performance of different MLRAs, and showed that the Gaussian processes regression (GPR), bagging trees (BAGTREE), and boosting trees (BOOST) are robust algorithms for LAI retrieval [13,25,26]. However, a drawback of MLRA methods is their instability when applied to data sets deviating from the training data set [25]. Therefore, additional investigation of such algorithms trained over a data set for crop-specific and mixed-crop estimation is required.

The third category includes the RTMs for the simulation of canopy light interception processes. Several authors have suggested the use of such complex models to exploit the full spectrum acquired with the RS sensors. Nevertheless, RTM calibration requires several input parameters, where the potential lack of these inputs could induce several uncertainties compromising the estimation accuracy [13].

Numerous LAI products have been developed, according to these different methodological approaches [14]. Gonsamo and Chen [27] used the MODIS/MISR sensor, with a parametric approach (LAI–VIs relationship), to obtained a global LAI at 250 m spatial resolution and 10 days temporal resolution. Yan et al. [28] used SNPP/VIIRS data retrieved a global LAI product at 8 days of temporal resolution and 500 m of spatial resolution by

RTM. Moreover, García-Haro et al. [29] used an AVHRR sensor to obtain an operational LAI product at 1.1 km of spatial resolution and 10 days of temporal resolution using GPR. Baret et al. [30] used a SPOT/VEGETATION sensor and, based on a hybrid non-parametric approach (Neural Network and GPR), retrieved a global LAI with 10 days of temporal resolution and 1.5 km of spatial resolution. Despite the large number of retrieved LAI products and the efforts to develop new and updated algorithms for LAI estimation, the available products are not yet capable of capturing the site-specific variability required in many agricultural applications, Therefore, specific LAI data sets with higher spatial and temporal consistency are required. To cope with the spatial and temporal variability of heterogeneous agricultural systems, high spatial resolution satellite systems (10–30 m), such as Sentinel-2, have been gathering growing interest in the field of agronomic research [24,26]. Thus, studies devoted to improving the accuracy and spatial resolution of LAI estimation are needed, in order to respond to the increasing demand for tools to support the site-specific management of crops and landscape [31]. Therefore, this study was focused on the robustness of non-parametric approaches in relation to different sources of variability such as crop species-, growth stage-, farm-, and year-specific conditions, in order to capture site-specific variability.

The objectives of this study were: (i) to evaluate the potential of non-parametric algorithms at pixel level (within field) and at field level for multi-crop and multi-temporal LAI retrieval; and (ii) to test the temporal consistency of the retrieved LAI at field level for crop monitoring over the entire growing season. Thus, to achieve these objectives, we performed an intensive field campaign, contemporary to S2 data acquisition, to collect ground-LAI measurements collected in Tuscany (Central Italy) over two growing seasons (2018 and 2019), including three crops (i.e., winter wheat, maize, and alfalfa), characterized by different growing periods and canopy structures, and considering different agronomic conditions (i.e., three farms in three different sites). Indeed, the final database (ground-LAI and S2 spectral data) is a potential contribution for other studies, thanks to the spatial–spectral–temporal characteristics of LAI data.

## 2. Materials and Methods

The methodology followed for the assessment of the performances of parametric and non-parametric approaches consisted of four steps: (i) data collection and pre-processing (Section 2.2); (ii) model definition parametric and non-parametric LAI retrieval (Section 2.3); (iii) accuracy assessment (Section 2.4); and (iv) temporal consistency analysis of LAI retrieval (Section 2.5). Concerning MLRAs, further analysis was performed, in order to evaluate the selection of bands in the retrieval process (Section 2.3.2.1). Indeed, the strength of MLRAs, with respect to regression based on VIs, involves the exploitation of the full spectral information of optical data. The spectral band contribution is important to assess such a contribution and to interpret the model results. The flowchart is presented in Figure 1.

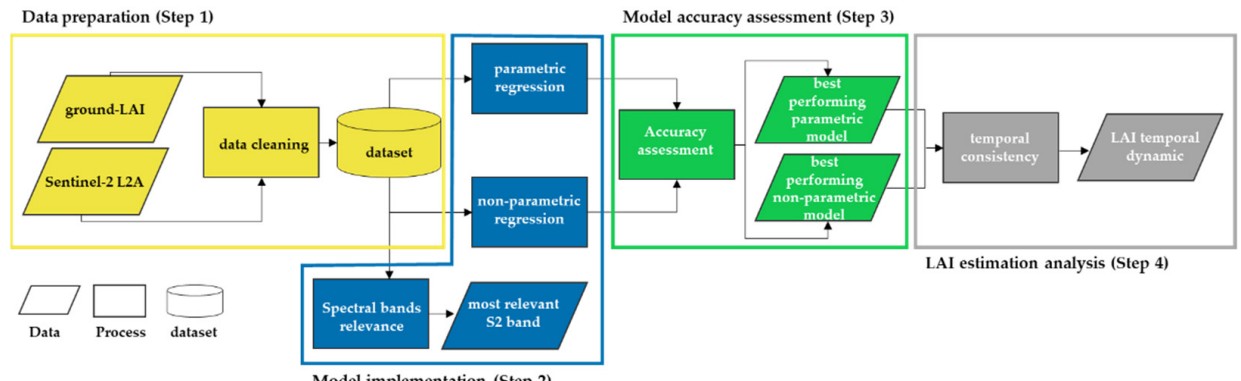

**Figure 1.** General framework of the work: Step 1 (Yellow boxes) data preparation; Step 2 (Blue boxes) model implementation; Step 3 (Green boxes) model accuracy assessment; and Step 4 (Grey boxes) LAI estimation analysis.

*2.1. Study Area*

The study area is located 10 km from Pisa, Tuscany, Central Italy (Figure 2a). The area is flat and the climate is Mediterranean, with a mean annual precipitation of 907 mm and a mean annual temperature of 15 °C (long-term average 1986–2016). According to land-cover spatial information of the Tuscany regional authorities (http://dati.toscana.it/, accessed on 20 April 2021), in 2017–2019, the three prevalent arable crop categories were: (i) cool-season cereals; (ii) perennial meadows; and (iii) warm-season cereals. In order to cope with the main categories of the area, the following crops were considered in this study: C1—winter wheat (*Triticum aestivum* L.); C2—maize (*Zea mays* L.); and C3—alfalfa (*Medicago sativa* L.). The three crops were monitored along their growing seasons at three different test sites. The three sites were located in three different Farms (F): F1 (San Piero), F2 (Coltano), and F3 (Madonna dell'Acqua), having an extension of the considered fields of about 25, 42, and 36 hectares, respectively (Figure 2b).

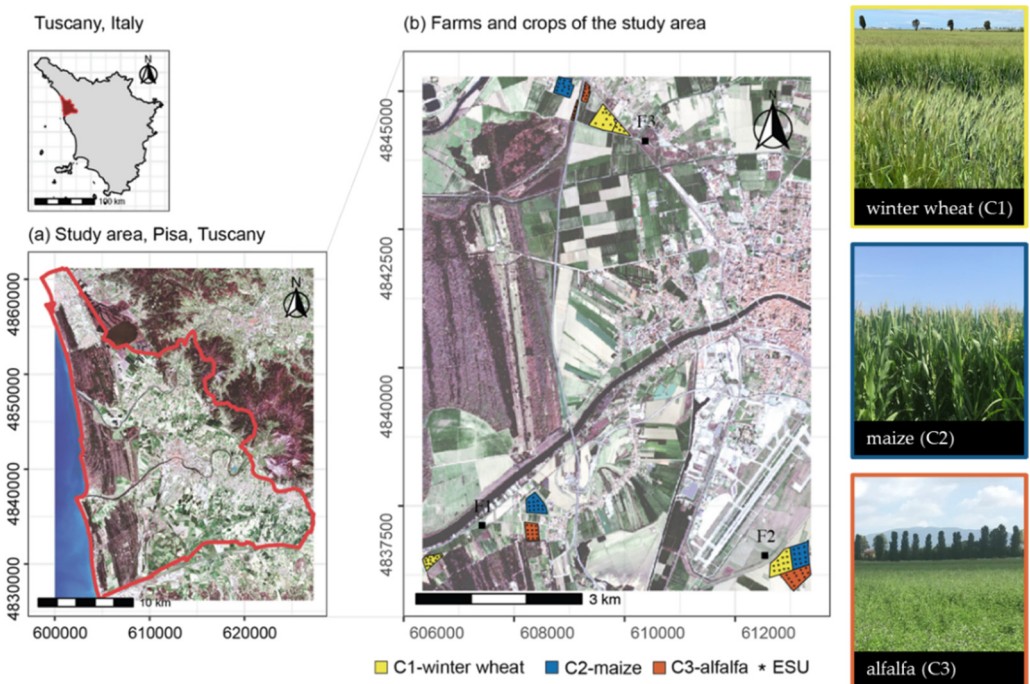

**Figure 2.** (**a**) Study area in Tuscany, Italy; (**b**) farms and crops of the study area (C1—winter wheat, C2—maize, and C3—alfalfa) in the three farms (F1—San Piero, F2—Coltano, and F3—Madonna dell'Acqua) in Pisa, Italy.

*2.2. Data*

2.2.1. In Situ Measurements

At each site, ground-LAI measurements were collected using the Validation of Land European Remote Sensing Instruments (VALERI) sampling strategy [32,33]. VALERI is based on an Elementary Sampling Units (ESU) upscaling approach, in order to capture the variability between the investigated fields and within the field of each crop, while including the variability within a theoretical pixel [33]. The corresponding scale of ESU is a high spatial resolution pixel of S2, and ESU is defined as a plot of 20 m × 20 m, where an investigated area of 1 m$^2$ was selected. The ground-LAI measurements were conducted from March 2018 to October 2019. In particular, in the 2018 field campaign, measurements were carried out monthly and in each farm; while, in the 2019 field campaign, weekly measurements were carried out only in F3. On each sampling date, for each field, we collected 12 ESU; where, within each ESU, four (2018) and one (2019) replicate measurements of ground-LAI were sampled. Subsequently, in order to obtain a unique representative value for ESU, the ground-LAI mean was calculated within each ESU. A

total of 598 measured ESUs were obtained over the study area. Ground-LAI measurements were performed using SunScan (Delta-T Devices, Cambridge, UK), assuming a random distribution of the foliage (effective ground-LAI) under clear-sky conditions. Moreover, to include bare soil conditions [34], 96 bare soil (LAI = 0.0) ESUs were included in the ground-LAI database (12 for each field, during the germination stage of winter wheat and maize for each sampling year). Therefore, we obtained a database consisting of 694 observations.

By means of phenological observations and expert knowledge, we conducted a preliminary screening of ground-LAI anomalies, in order to exclude sampling disturbances arising out of instrument calibration. Moreover, Grubbs' test was performed on ground-LAI measurements data, in order to identify and flag outliers [35]. Both screenings were conducted per field per date, in order to avoid possible sources of disturbance on the day of sampling. Grubbs' test was performed with the "outliers" library in the R environment [36]. After the screenings, a total of 558 representative ESUs were maintained, while 232 ESUs were excluded from the measured ground-LAI database (Table 1).

**Table 1.** Number of ESUs sampled in the three farms (F1, F2, and F3) during the 2018 and 2019 field campaigns for the three crops (C1 = winter wheat; C2 = maize; C3 = alfalfa).

| Year of Sampling Campaign | Farm | Crop | ESU |
|---|---|---|---|
| 2018 | F1 | C1 | 55 |
| | | C2 | 48 |
| | | C3 | 48 |
| | F2 | C1 | 33 |
| | | C2 | 36 |
| | | C3 | 25 |
| | F3 | C1 | 26 |
| | | C2 | 48 |
| | | C3 | 31 |
| 2019 | F3 | C1 | 56 |
| | | C2 | 72 |
| | | C3 | 80 |
| Total | | | 558 |

Beside LAI measurements, the phenological stages of the crops were recorded during each sampling date, based on the Biologische Bundesanstalt, Bundessortenamt, and Chemical Industry scale (BBCH), a German scale used to identify the phenological development stages of a plant [37]. Data of crop phenology were arranged, in order to refer LAI measurements to the main crop development stages. Thus, for winter wheat and maize, four main stages were identified: GE, from germination to full emergence; SE, from early leaf development to complete stem elongation; Fl, from initial stages of flower differentiation to end of anthesis; and FD, from initial stages of fruit development to complete fruit maturity. Conversely, in alfalfa, only two stages were considered: Vg, the vegetative stage, occurring after cut or overwintering; and Fl, from initial stages of flowering to mowing.

### 2.2.2. Sentinel-2 Data

Copernicus Sentinel-2 (S2) is a satellite mission carrying the MSI multispectral sensor, which is characterised by high spatial resolution (10 m, 20 m, and 60 m), high revisit capability (5 days with two satellites), and a moderately large band set (13 spectral bands) from the visible to short-wave infrared (Table A1) [38,39]. The S2 Level 2A (L2A) images were downloaded from the Theia Land Data Centre, which provides time-series of top-canopy surface reflectance which is orthorectified, terrain-flattened, and atmospherically corrected using the MACCS-ATCOR Joint Algorithm (MAJA) [40]. A total of 37 cloud-free images, collected in correspondence with the in situ monitoring period, were used to analyze the relationship between measured ground-LAI and S2 data. Moreover, the most

commonly-used S2 bands for vegetation studies at 10 m (B02, B03, B04, and B08) and 20 m (B05, B06, B07, B08A, B11, and B12) were selected for the analysis (Table A1) [31]. Then, the 10 m spatial resolution bands were resampled to 20 m spatial resolution.

In order to couple the ground-LAI values and the S2 spectral information, the centroid of each ESU was used to extract zonal statistics of S2 time-series using the R software package ''raster'' [36,41]. S2 reflectance data values within ±5 days from ground data collection were associated to ground-LAI values. The association of ground-LAI values with the corresponding S2 data was carried out using the SQL database software PostgreSQL 9.5, by joining the two data sets. As a result, a complete SQL database, consisting of 558 records of coupled ground-LAI and S2 data, was obtained for the three crops of each farm in the reference period (October 2017 to October 2019; Figure A1 (Appendix A)).

### 2.3. LAI Retrieval Approaches

#### 2.3.1. Parametric Methods

By means of a parametric approach, the empirical relationship between vegetation indices (VIs) and ground-LAI was analyzed, including different sources of variability (crop type, farm conditions, and crop growth). The VIs given in Table 2 were selected, based on previous studies which evaluated visible, red edge, and shortwave infrared as the most effective wavelengths for LAI estimation over different crop types [34,42].

**Table 2.** Vegetation Indices (VIs).

| VIs | Name | Formula | References |
|---|---|---|---|
| NDVI | Normalized difference vegetation index | (B08-B04)/(B08 + B04) | [43] |
| SeLI | Simple Sentinel-2 LAI Index | (B8A-B05)/(B08A + B05) | [34] |
| NBR | Normalized Burn Ratio | (B08-B12)/(B08 + B12) | [44] |

Linear and non-linear functions were used for LAI prediction. The linear model (LM) was selected, according to a previous study which demonstrated the good performance of LM for LAI prediction in a mixed-crop scenario [34]. The non-linear model was selected, considering the results of previous studies evaluating the relationship between the ground-LAI and VIs [42,45]. Specifically, the dose–response three-parameter logistic model was selected, in order to evaluate the logistic relationship between ground-LAI and VIs [21]. For the linear model (LM), we adopted the general function of Equation (1) and its inverse (Equation (2)), where $a$ is the intercept and $b$ is the slope:

$$VI = a + b\,\text{groundLAI}, \qquad (1)$$

$$LAI = \frac{(VI - a)}{b}. \qquad (2)$$

The inverse non-linear function ($\text{LogIF}_d$) of the three-parameter logistic model (Equation (3)) was computed based on the parameterization carried out by the dose–response "drc" R package [46]. In the equation, $d$ is the curve plateau, which indicates the level at which the VIs saturate; $b$ is the relative slope, which indicates the steepness of the curve; and $e$ is the inflection point, which indicates the point at which the VI value is halfway to its saturation level. Based on Equation (3), the inverse of the three-parameter logistic model was calculated using Equation (4).

$$VI = \frac{d}{\{1 + \exp[b * (\text{groundLAI} - e)]\}}, \qquad (3)$$

$$LAI = \ln(d/VI - 1)/b + e. \qquad (4)$$

Considering that the domain of the inverse function is $VI \leq d$, the LAI results are unpredictable above the asymptote $d$, as the function is not real. In order to overcome the problem of losing information due to the saturation of VIs, we evaluated the option

of assigning a default value to pixels with VI > *d*, which is equal to the maximum value of VI measured below *d*; while, for VI ≤ *d*, the pixel value was computed based on the inverse function.

Moreover, to evaluate the robustness of the LM and LogIF$_d$ over different sources of variability, the functions were parametrized according to the mixed-crop (MC) data set and the crop-specific (CS) subset. Regression analysis was performed using the "drc" package [47,48] in the R environment [36]. In particular, the general model fitting function dose–response model (drm) was used to fit the regression models and, according to the same parameterization, the inverse functions were calculated.

### 2.3.2. Non-Parametric Methods

The MLRAs were used for multi-crop and multi-temporal LAI estimation by a non-parametric approach. The architecture of these MLRAs and their training processes were based on iterative regression process, and the core of each algorithm is reported in Table 3. More specifically, GPR provides the predictive mean, as well as predictive variance, maximizing the marginal likelihood in the training set, which is learned by hyperparameters through an appropriate kernel function [49]. BAGTREE builds multiple decision trees by iteratively replacing resampled training data and voting for the decision trees, thus leading to a consensus prediction [50]. BOOST incrementally builds an ensemble by training each new instance to emphasize the training instances which were previously mismodelled [51]. Moreover, to evaluate the MLRA performances over different sources of variability, the algorithms were trained over the mixed-crop (MC) and crop-specific (CS) data set collected during the 2018 and 2019 field campaigns. The MLRAs were run by means of the MLRA toolbox [52] of the Automated Radiative Transfer Models Operator (ARTMO) software [53].

**Table 3.** Machine learning regression algorithms (MLRAs) used in this study.

| MLRAs | Code | Methods |
|---|---|---|
| Bagging trees | BAGTREE | Bootstrap method and regression trees |
| Gaussian Process Regression | GPR | Bayesian statistical inference |
| Boosting trees | BOOST | Least squares boosting and regression trees |

### 2.3.2.1. Spectral Band Relevance

Crop development stages strongly influence the reflectance information provided by RS sensors. Therefore, information on spectral relevance may be a useful tool for understanding several crop-related reflectance conditions. In this study, the spectral relevance of each S2 band (Section 2.2.2) used for LAI prediction was evaluated, according to the crop type and crop development stage. The GPR models were trained over each growth stage within the crop-specific data sets (CS) and over the crops on the mixed-crop data set (MC). We exploited the property of GPR to evaluate the predictive capacity of each single band by a covariance function, defined as follows:

$$K\left(x_i,\ x_j\right) = \exp\left(-\frac{\parallel x_i,\ x_j \parallel^2}{2\sigma^2}\right). \tag{5}$$

Information of spectral relevance of each S2 band was obtained by the hyperparameter sigma ($\sigma$), which is the weight assigned to the band (*b*). The $\sigma_b$ value was provided by the GPR within the MLRA toolbox implemented in the ARTMO software [52]. The toolbox provides an absolute $\sigma_b$ value for each used band. In the supplied hyperparameter, the higher the value of $\sigma_b$, the lower the relevance of the band. In this study, in order to obtain a positive representation of the spectral relevance, we converted the lower values of $\sigma_b$ into higher ones and calculated the relative $\sigma_b$. This is detailed in Equation (6), where $\sigma_b$ is the

spectral relevance, max$\sigma_b$ is the maximum value of $\sigma_b$ among the bands used for the GRP training, and sum$\sigma_b$ is the sum of $\sigma_b$ among the bands used for the GRP training:

$$\sigma^2 = \left(1 - \left(\sigma^2 / \left(\max\sigma^2\right)\right) / \left(\text{sum}\sigma^2\right)\right) * 100 \tag{6}$$

*2.4. Accuracy Assessment*

The accuracy of parametric and non-parametric methods was assessed at pixel and field level using K-fold cross-validation. The K-fold process consists of dividing the data set into k mutually exclusive groups following a k-fold cross-validation partitioning design [54]. In our case, data were randomly split into $k = 3$ subsets of equal size, of which, iteratively, two were used for calibration and one for validation. Using this approach, the dependence on a single random partition into calibration and validation data sets was reduced and all observations were used for both training and validation, with each observation used for validation just one time [55]. The validation at field level was conducted using linear regression between ground-LAI values and predicted LAI, averaged per crop, per field, and per growing stage, while the validation at pixel level was conducted at the ESU scale. The coefficient of determination ($R^2$) and root mean square error (RMSE) were calculated to assess the prediction accuracy. The parametric and non-parametric models were selected as those robust and efficient for LAI prediction, considering the average value of RMSE from the cross-validation (RMSE$_{CV}$) and the average of coefficient of determination estimated between ground-LAI and predicted LAI ($R^2_{CV}$). Finally, the Wilcoxon signed-rank test was used to compare the distributions of predicted LAI values, according to the crop-specific (CS) and mixed-crop (MC) parameterization, and to identify statistical differences between the two parameterization approaches.

*2.5. Temporal Consistency*

For assessment of the temporal consistency of the estimated LAI against the measured LAI, the 2019 ground-LAI data set was used. First, the LAI time-series from the Sentinel-2 L2A data from October 2018 to October 2019 were retrieved, in order to cover the entire reference period. Then, the temporal consistency was evaluated.

To assess the temporal consistency of the predicted LAI for the temporal profiles, one representative ESU per crop was randomly selected. Subsequently, in order to obtain a full time-series, the ESU profile was interpolated and smoothed using the GPR method. Then, profiles of the retrieved LAI were compared with the measured ground-LAI and the related standard errors. The temporal profile was filtered using the Decomposition and Analysis of Time Series software (DATimeS) [56].

## 3. Results

*3.1. Seasonal Variation of Ground-LAI*

During the two growing seasons (i.e., 2018 and 2019), different ranges of ground-LAI values were measured within the three farms for each crop type (Figure 3).

During 2018, for winter wheat (C1) (Figure 3a), the minimum ($0.75 \pm 0.11$) and maximum ($4.67 \pm 0.95$) values of ground-LAI were observed at the San Piero farm (F1), at DOY 84 and 109, respectively. Meanwhile, in 2019 (Figure 3b), the minimum ($1.52 \pm 0.33$) and maximum ($5.54 \pm 0.68$) ground-LAI values of C1 were observed at DOY 56 and DOY 106 at the Coltano farm (F3), respectively.

Regarding maize (C2) (Figure 3c,d), the minimum ground-LAI value for 2018 ($1.10 \pm 0.48$) was observed at DOY 208, while that for 2019 was observed at DOY 190 ($1.32 \pm 0.50$), while the maxima for 2018 ($5.21 \pm 2.52$) and 2019 ($4.37 \pm 0.83$) were at DOY 241 and 248, respectively.

For alfalfa (C3), in 2018 (Figure 3e) the minimum ground-LAI was 0.88 ($\pm 0.26$) at F1 (DOY 163) and the maximum was 9.41 ($\pm 0.80$) at F3 (DOY 263). During 2019 (Figure 3f), at F3 the ground-LAI ranged between 1.64 ($\pm 0.40$) and 5.10 ($\pm 0.59$) at DOY 191 and 106, respectively.

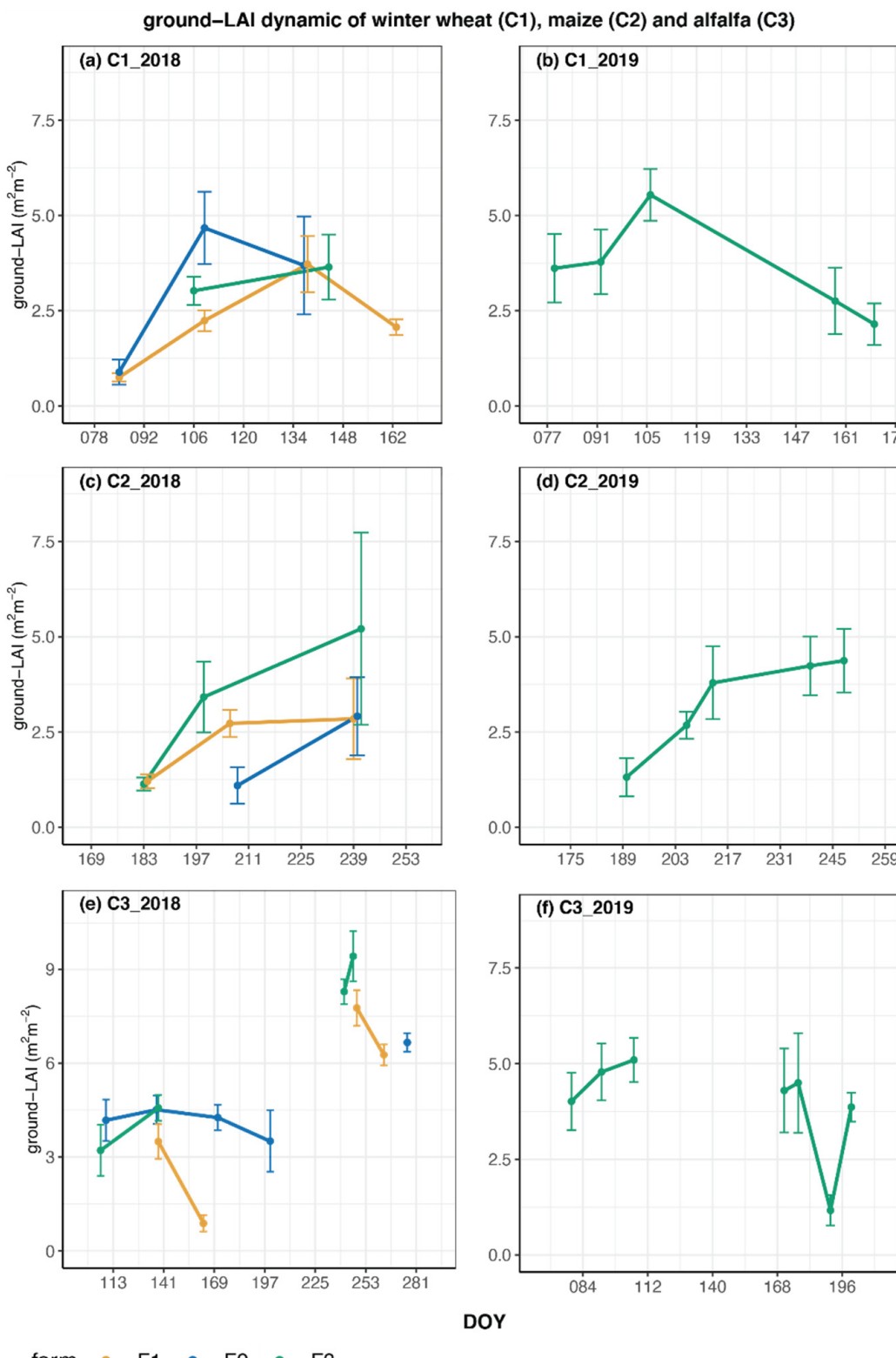

**Figure 3.** Mean values of the measured ground-LAI values of winter wheat (C1), maize (C2), and alfalfa (C3), according to the main crop development stages. The growing stages for winter wheat and maize are: SE, stem elongation; Fl, flowering; and FD, fruit development. For alfalfa, the growing stages are: Vg, vegetative; Fl, flowering.

*3.2. Model Implementation and Accuracy*

3.2.1. Parametric Model

To evaluate the predictability of LAI by VIs (NDVI, SeLI, and NBR), the accuracy of the three-parameter logistic ($LogIF_d$) and linear (LM) functions were assessed over the crop-specific (CS) and mixed-crop (MC) parameterizations. Moreover, the LM and $LogIF_d$ MC parameterizations were assessed and compared at pixel and field level, in order to assess the suitability of VIs for the prediction of LAI in a mixed-crop scenario.

Table 4 shows the results of the accuracy of the parametric functions at pixel level. In general, it was observed that the LM showed statistical differences between the predicted LAI values for both mixed-crop and crop-specific parameterizations ($p$-value < 0.05). In contrast, $LogIF_d$ showed that there were no statistical differences between the two parameterizations based on NBR for winter wheat (C1), NDVI for maize (C2), and SeLI for alfalfa (C3) ($p$-value > 0.05). Overall, the two parametric methods showed $R^2_{CV}$ values lower than 0.73 for the three VIs. For the LM, the coefficient of determinations were similar under both the CS and MC parameterizations, with a lower $RMSE_{CV}$ under CS parameterization in all the three crops for each VI. Conversely, the $R^2_{CV}$ and $RMSE_{CV}$ of the $LogIF_d$ model showed differences between mixed-crop and crop-specific parameterizations for all three crops. Both of the parametric methods showed low accuracy in C3, regardless of the VIs and the parameterization data set. With the LM, the strongest accuracy of predicted LAI values of C1 was obtained using NBR with the CS parameterization ($R^2_{CV}$ = 0.72; $RMSE_{CV}$ = 0.68) and using SeLI with the MC parameterization ($R^2_{CV}$ = 0.72; $RMSE_{CV}$ = 0.73). Instead, when using the $LogIF_d$ in C1, NBR showed the highest accuracy in both the cases of CS ($R^2_{CV}$ = 0.72; $RMSE_{CV}$ = 0.98) and MC ($R^2_{CV}$ = 0.73; $RMSE_{CV}$ = 0.75) parameterizations.

**Table 4.** Cross-validation results of LAI estimation at pixel level from SeLI, NDVI, and NBR using the linear function (LM) and the inverse function improved with the conditional statement d ($LogIF_d$). The table reports, for each crop (C1 = winter wheat, C2 = maize, and C3 = alfalfa), the metrics obtained, according to the parameterization made on the crop-specific (CS) and the mixed-crop (MC) data sets. The table reports the mean of coefficient of determination ($R^2_{CV}$) and root mean square error ($RMSE_{CV}$) estimated from the k-fold cross validation procedure ($k$ = 3); as well as the slope and the intercept of regression lines. The $p$-values < 0.05 (*), <0.01 (**), and <0.001 (***) indicate significant differences between the predicted LAI with MC and CS data sets.

| Function | Metrics | Data Set | C1 | | | C2 | | | C3 | | |
|---|---|---|---|---|---|---|---|---|---|---|---|
| | | | NBR | NDVI | SeLI | NBR | NDVI | SeLI | NBR | NDVI | SeLI |
| LM | $R_{CV}^2$ | CS | 0.73 | 0.72 | 0.73 | 0.66 | 0.64 | 0.65 | 0.08 | 0.14 | 0.17 |
| | | MC | 0.73 | 0.72 | 0.73 | 0.65 | 0.64 | 0.65 | 0.08 | 0.14 | 0.17 |
| | $RMSE_{CV}$ | CS | 0.68 | 0.73 | 0.72 | 0.81 | 0.89 | 0.88 | 0.67 | 0.80 | 0.84 |
| | | MC | 0.76 | 0.83 | 0.72 | 0.94 | 1.07 | 1.12 | 0.99 | 0.96 | 0.87 |
| | Intercept | CS | 0.65 | 0.54 | 0.52 | 0.98 | 0.77 | 0.77 | 4.15 | 3.92 | 3.83 |
| | | MC | 1.17 | 0.80 | 0.81 | 1.21 | 0.92 | 1.12 | 2.85 | 2.87 | 2.72 |
| | Slope | CS | 0.69 | 0.72 | 0.73 | 0.60 | 0.64 | 0.65 | 0.10 | 0.15 | 0.17 |
| | | MC | 0.76 | 0.82 | 0.73 | 0.69 | 0.78 | 0.82 | 0.15 | 0.18 | 0.18 |
| | $p$-value | CS vs. MC | 0.00 | 0.00 | 0.00 | 0.00 | 0.00 | 0.00 | 0.00 | 0.00 | 0.00 |
| $LogIF_d$ | $R_{CV}^2$ | CS | 0.72 | 0.70 | 0.62 | 0.60 | 0.59 | 0.55 | 0.05 | 0.04 | 0.09 |
| | | MC | 0.73 | 0.72 | 0.65 | 0.64 | 0.62 | 0.65 | 0.02 | 0.03 | 0.05 |
| | $RMSE_{CV}$ | CS | 0.98 | 1.09 | 1.33 | 1.00 | 1.14 | 1.31 | 0.84 | 1.27 | 1.46 |
| | | MC | 0.75 | 0.86 | 0.85 | 0.93 | 1.16 | 1.23 | 1.02 | 1.32 | 1.41 |
| | Intercept | CS | 0.23 | 0.20 | 0.16 | 0.55 | 0.60 | 0.60 | 2.13 | 2.53 | 2.43 |
| | | MC | 0.43 | 0.29 | 0.25 | 0.50 | 0.54 | 0.58 | 2.39 | 2.71 | 2.61 |
| | Slope | CS | 0.97 | 1.03 | 1.05 | 0.65 | 0.71 | 0.76 | 0.11 | 0.16 | 0.22 |
| | | MC | 0.77 | 0.85 | 0.73 | 0.67 | 0.78 | 0.89 | 0.09 | 0.15 | 0.15 |
| | $p$-value | CS vs. MC | 0.51 | 0.00 | 0.00 | 0.01 | 0.60 | 0.24 | 0.00 | 0.00 | 0.14 |

With the LM function in C2, the highest precision was obtained by NBR for both CS ($R^2_{CV}$ = 0.66; $RMSE_{CV}$ = 0.81) and MC ($R^2_{CV}$ = 0.65; $RMSE_{CV}$ = 0.94) parameterizations. Regarding the case of the $LogIF_d$ function, instead, the highest accuracy was obtained by SeLI for the CS parameterization ($R^2_{CV}$ = 0.65; $RMSE_{CV}$ = 1.23) and by NBR ($R^2_{CV}$ = 0.60; $RMSE_{CV}$ = 0.93) for the MC parameterization. Conducting a comparison among all three of the crops, LM exhibited the weakest accuracy in C3, considering all the VIs, with a coefficient of regression lower than 0.17. Analyzing the slope and the intercept of the regression equation, it can be argued that the LM led to overestimation, particularly in the case of low LAI values, independent of the VIs and the parameterization data set used. On the other hand, $LogIF_d$ strongly overpredicted when the LAI value was in the high–medium range.

In C1, when using SeLI in both parameterizations, the intercepts showed low values (CS = 0.52 and MC = 0.81 for LM; CS = 0.16 and MC = 0.25 for $LogIF_d$); $LogIF_d$ even exhibited values strictly close to 0. At the same time, in the presence of medium–high LAI values, the slope of SeLI also tended to overpredict under the CS parameterization with a value of 1.05 and underpredicted for MC parameterization with a value of 0.73.

In the case of C2, a completely different performance was obtained, according to the two functions and the three VIs; in fact, in the case of LM, the lowest intercept was obtained with NDVI for both parameterizations (MC = 0.77; CS = 0.92). In the $LogIF_d$ case, the lowest intercept was obtained with NBR for both parameterizations (MC = 0.55; CS = 0.50).

At the field level, both parameterizations of LM (Figure A5) and $LogIF_d$ (Figure 4) showed an $R^2$ value higher than 0.8 for both C1 and C2. When LM was used, in both C1 and C2, the highest accuracy for the MC parameterization was obtained with SeLI, whose $R^2_{CV}$ was equal to 0.88 and 0.82 for C1 and C2, respectively. Otherwise, for C1 and C2 in the MC parameterization, $LogIF_d$ showed a higher performance using NBR, presenting $R^2_{CV}$ values equal to 0.87 and 0.85, respectively.

### 3.2.2. Non-Parametric Model

Even for non-parametric models, MLRAs were trained over the crop-specific (CS) and mixed-crop (MC) data sets, in order to evaluate and compare the accuracy of LAI prediction at pixel level. Table 5 shows the accuracy metrics of MLRAs at the pixel level. In general, the GPR showed the highest $R^2_{CV}$ and the lowest $RMSE_{CV}$ for each training data set of all the three crops, except for C2. In fact, when BAGTREE was used in the mixed-crop data set of C2, the $R^2_{CV}$ was 0.62, compared to 0.59 under GPR, and the $RMSE_{CV}$ was 1.02 vs. 1.06 of GPR. When GPR was used in C3, the estimated values were slightly overestimated; indeed, the values of the intercept higher than 1 confirmed this overprediction. However, despite the overestimation of low-LAI values with GPR, the slope values close to 1 highlighted that there was a positive linear relationship between the measured and predicted values.

Overall, the MLRAs showed a higher accuracy of prediction at pixel level than parametric methods, and it was also observed that GPR did not provide any significant difference between the predicted LAI values according to the MC and CS data sets, regardless of the considered crop. The higher efficacy of MRLAs, in terms of the estimation of LAI under both the MC and CS data sets, was also observed at the field level (Figure 5). Indeed, GPR obtained an $R^2_{CV}$ value higher than 0.80 for all the crops.

### GPR Spectral Band Relevance

The graphs shown in Figure 6 synthesize the analysis performed to evaluate the relevance of S2 spectral bands to the different GPR models for LAI estimation, according to crop development stages for the "crop-specific model" and crop type for the "mixed-crop model".

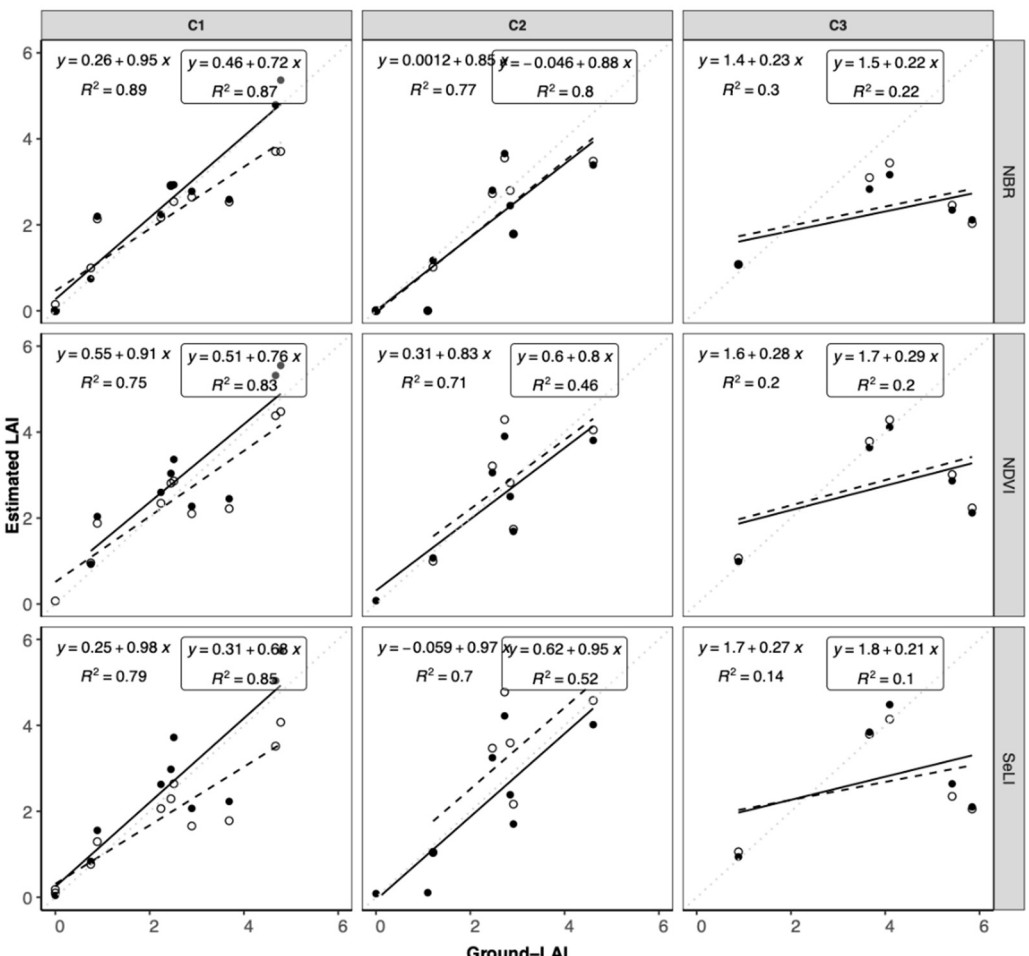

**Figure 4.** Validation results of LAI estimation at field level from SeLI, NDVI, and NBR by the inverse function improved with the conditional statement d (LogIF$_d$). Results per crop (C1, C2, and C3) of linear regression analysis between measured ground-LAI and LAI predicted from NDVI, SeLI, and NBR by the mixed-crop (MC) LogIF$_d$ (dashed line on black dots and equation in the box) and the crop-specific (CS) LogIF$_d$ (continuous line on white dots).

**Table 5.** Cross-validation results of LAI estimation at pixel level from GPR, BOOST, and BAGTREE. The table reports, for each crop (C1 = winter wheat, C2 = maize, and C3 = alfalfa), the metrics obtained according to the parameterization made on the crop-specific (CS) and the mixed-crop (MC) data sets. The table reports the mean of coefficient of determination ($R^2_{CV}$) and root mean square error (RMSE$_{CV}$) estimated from the k-fold cross validation procedure (k = 3); as well as the slope and the intercept of regression lines. *p*-values < 0.05 (*), < 0.01 (**), and < 0.001 (***) indicate significant differences between the predicted LAI with MC and CS data sets.

| Metrics | Dataset | C1 | | | C2 | | | C3 | | |
|---|---|---|---|---|---|---|---|---|---|---|
| | | GPR | BOOST | BAGTREE | GPR | BOOST | BAGTREE | GPR | BOOST | BAGTREE |
| $R_{CV}^2$ | CS | 0.81 | 0.79 | 0.77 | 0.69 | 0.63 | 0.67 | 0.71 | 0.62 | 0.63 |
| | MC | 0.77 | 0.63 | 0.74 | 0.59 | 0.58 | 0.62 | 0.65 | 0.62 | 0.62 |
| RMSE$_{CV}$ | CS | 0.65 | 0.68 | 0.66 | 0.84 | 1.04 | 0.87 | 1.00 | 1.22 | 1.04 |
| | MC | 0.66 | 0.99 | 0.77 | 1.06 | 1.15 | 1.02 | 1.05 | 1.07 | 1.03 |
| Intercept | CS | 0.36 | 0.29 | 0.45 | 0.67 | 0.56 | 0.67 | 1.36 | 1.26 | 1.65 |
| | MC | 0.51 | 0.43 | 0.55 | 0.72 | 0.77 | 0.75 | 1.42 | 1.61 | 1.52 |
| Slope | CS | 0.82 | 0.81 | 0.75 | 0.68 | 0.73 | 0.68 | 0.71 | 0.71 | 0.63 |
| | MC | 0.75 | 0.81 | 0.80 | 0.69 | 0.72 | 0.70 | 0.65 | 0.61 | 0.59 |
| *p*-value | CS vs. MC | 0.13 | 0.08 | 0.00 | 0.30 | 0.06 | 0.06 | 0.00 | 0.27 | 0.00 |

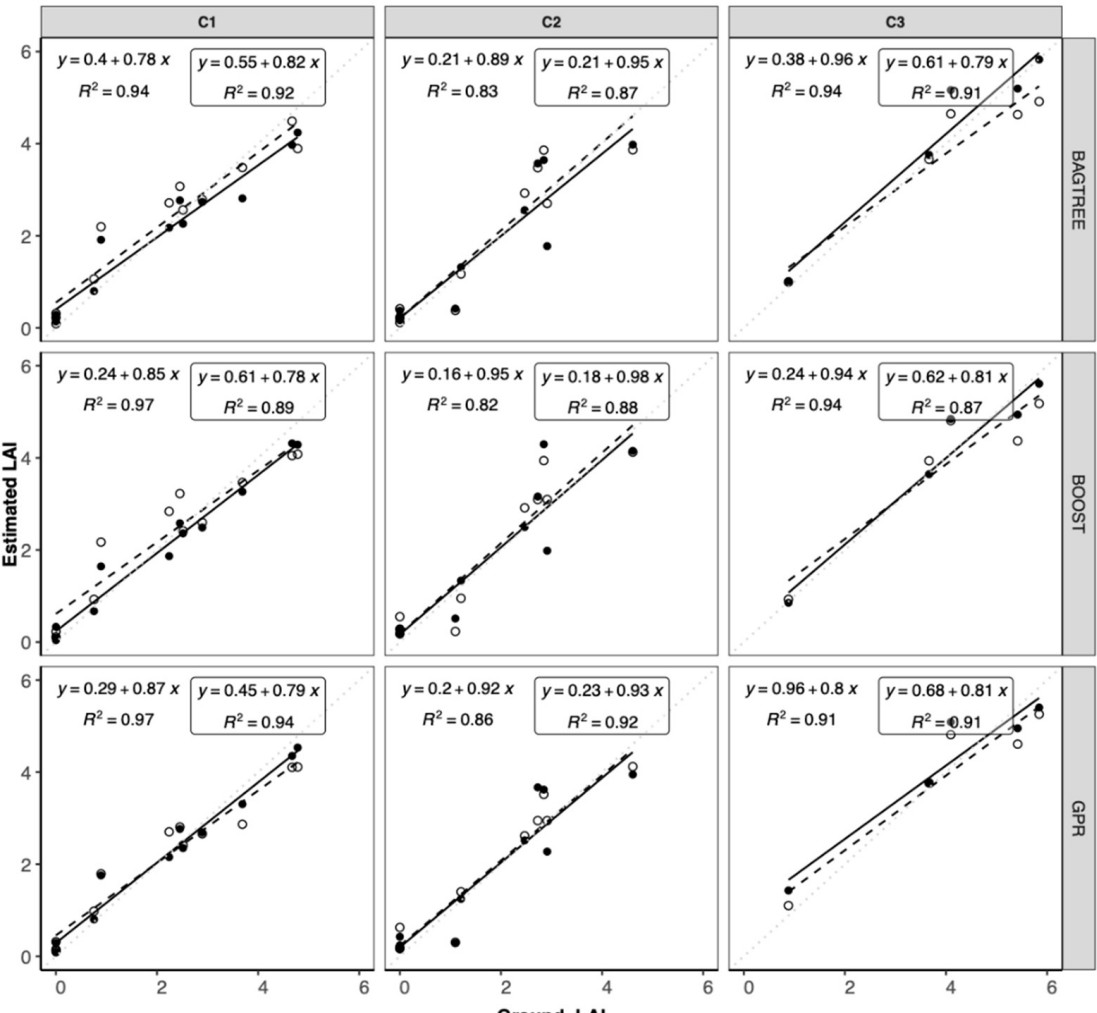

**Figure 5.** Validation results of LAI estimation at field level by GPR, BOOST, and BAGTREE. Results per crop (C1, C2, and C3) of the linear regression analysis between measured ground-LAI and LAI predicted from GPR, BOOST, and BAGTREE, according to the mixed-crop (MC) (dashed line on black dot and equation in the box) and crop-specific (CS) (continuous line on white dots) training data sets.

Regarding the crop-specific model, when C1 (winter wheat) (Figure 6a) was in the stem elongation (SE) stage, the highest relevant bands were in the red edge (RE) region, as B5 and B6 showed a spectral contribution of 22% and 23%, respectively, for a total of more than 45%. Conversely, during the flowering (Fl) stage, a slight decrease of RE contribution (15% B5 and 15% B6), and a significant increase in the near-infrared (NIR) region (B7, B8, and B8A), higher than 15%, was observed. When C1 reached the fruit development (FD) stage, all bands in the VIS-NIR region had a similar relevance, ranging from 10% to 13% contribution, while those in the SWIR region (B11 and B12) showed values lower than 2%.

When C2 (maize) (Figure 6b) was in the SE stage, the highest contribution was observed in the NIR (B7, B8, and B8A) and RE regions (B5 and B6), which reached a relevance higher than 10% (up to 14%), while the lowest contribution was exhibited by the blue band (B2), with 0% relevance. During the Fl stage of C2, B8A (with about 14%) showed the highest percentage of relevance, followed by B7 and B8 (with more than 12%). During the FD stage of C2, B8 showed the highest percentage of relevance (18%), B2 also had a great contribution (17%), as well as B7 and B6 (both about 14%), while the contributions of other bands were all around or less than 5%.

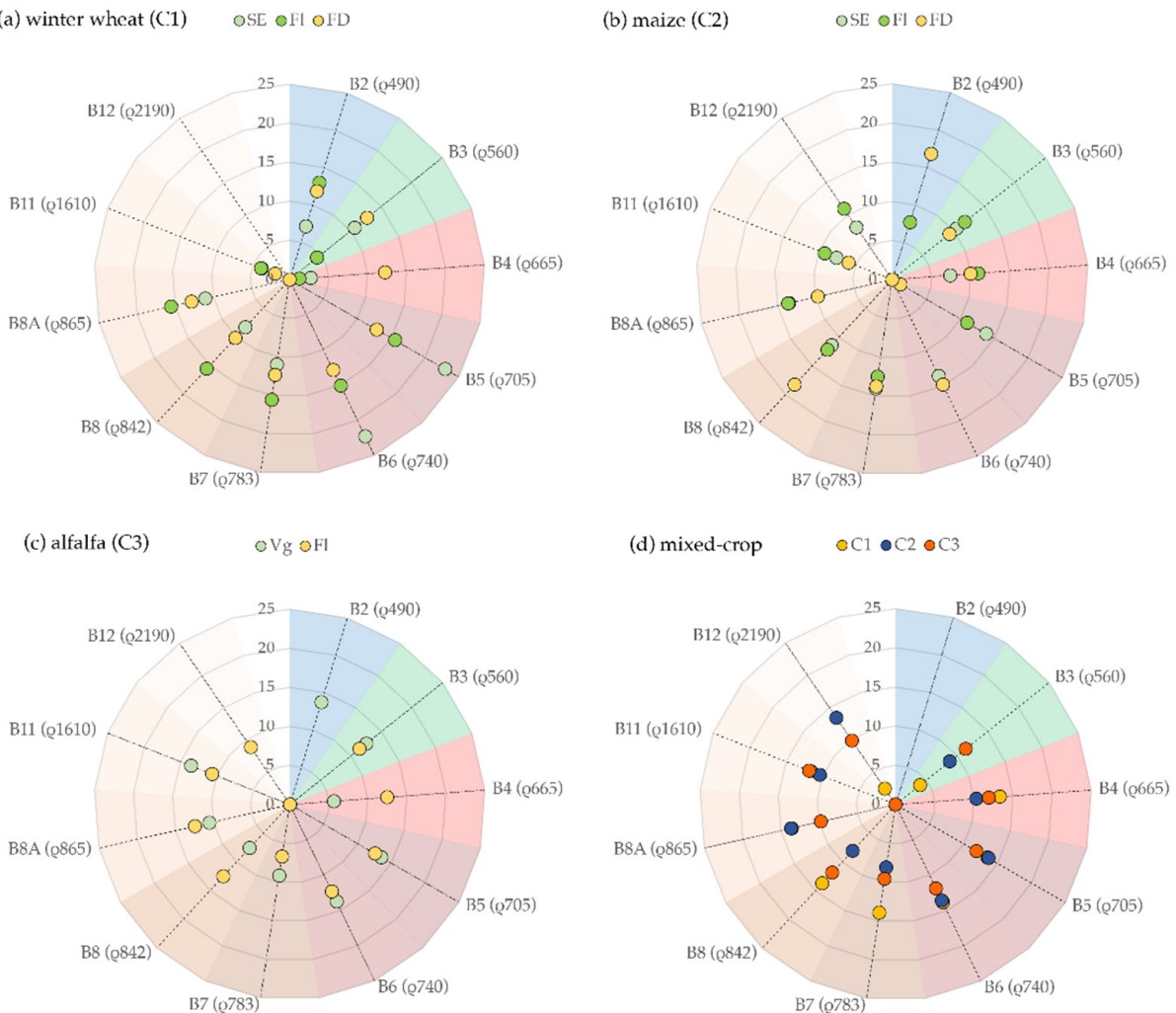

**Figure 6.** Radar plots of the relative weight (%), estimated from σ values provided by GPR analysis, of Sentinel-2 spectral bands to LAI estimation, according to: (i) different growing stages within the crop-specific data set for C1 (winter wheat), C2 (maize), and alfalfa (C3); and (ii) crop type within the mixed-crop data set. The growing stages for winter wheat and maize are: GE, from germination to emergence; SE, stem elongation; Fl, flowering; and FD, fruit development. For alfalfa, the growing stages were: Vg, vegetative; and Fl, flowering.

Apart from B2 (0%) and B7 (7%), when C3 (alfalfa) (Figure 6c) was in the vegetative stage (Vg), a uniform spectral contribution was observed across all of the spectrum, with a relevance higher than 10%. When C3 was in the Fl stage, the lowest percentage was observed with respect to B4 (6%) and the highest was with B6 (14%).

Analysis of the mixed-crop data set (Figure 6d) by GPR showed that the RE and NIR regions provided the largest contribution (about 13% for the spectral bands B5, B6, B7, B8, and B8A) for C1 LAI estimation. Regarding C2 LAI estimation, the most relevant bands were B6, B7, and B8A, with percentage higher than 13%. In C3, the LAI model exploited all of the spectrum, with a similar contribution (higher than 10%); except for B2, which was not relevant (0%).

### 3.3. Temporal Consistency of LAI estimation

Using the data of the 2019 sampling campaign, the parametric function and non-parametric algorithm with the highest accuracy for LAI retrieval were selected for the temporal assessment. Figure 7 show the dynamics of the predicted LAI values compared with the measured ground-LAI of one representative ESU for each crop by LM, LogIF$_d$, and GPR. In general, the predicted LAI profiles obtained from the GPR exhibited the closest

predicted LAI values to the measured ground-LAI, while the parametric approaches based on LM and LogIF$_d$ showed different consistency of temporal dynamics, according to the crop type.

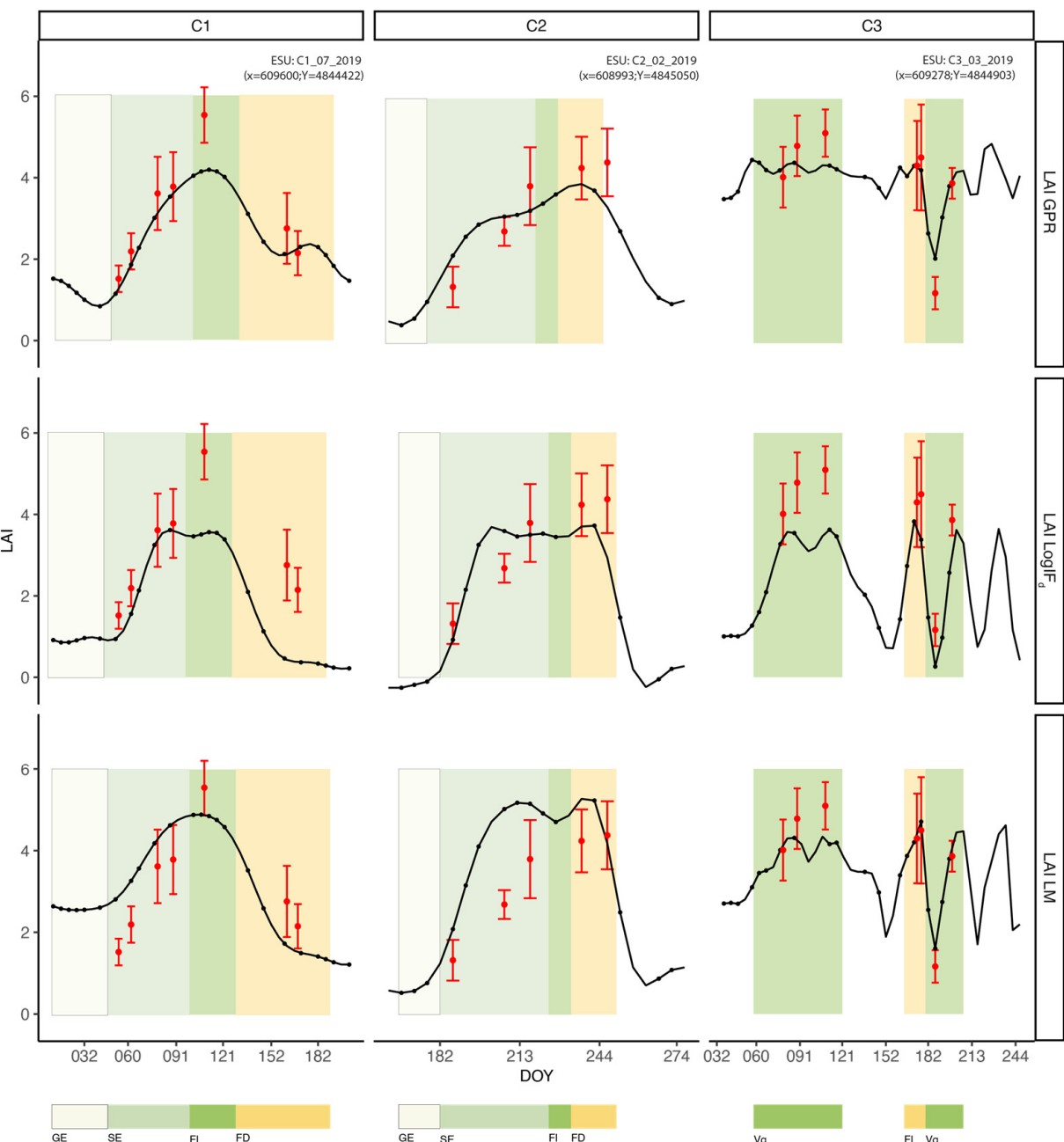

**Figure 7.** Temporal profile of LAI retrieved with GPR, LM, and LogIF$_d$ (black line), compared with the measured ground-LAI (red dots). Black lines represent the smoothed temporal profile of each crop using the GPR interpolation method, while black dots are the available S2 images. Red bars represent the standard deviation of ground-LAI. Different colored areas indicate the duration of each growth stage. In winter wheat (C1) and maize (C2), the growth stages are: from germination to full emergence (GE); stem elongation (SE); flowering (Fl); and fruit development (FD). In alfalfa (C3), the growth stages are: vegetative (Vg), occurring after mowing and flowering; and (Fl) occurring at full canopy development before mowing. In the *x*-axis, the day of the year (DOY) is depicted.

Specifically, it was observed that LogIF$_d$ and GPR in the stem elongation (SE) stage of winter wheat (C1) showed a similar behavior, yielding predictions very close to the ground-LAI values. Conversely, the LM exhibited the most distant predictions to the measured

LAI. In C1 flowering (Fl) and fruit development (FD) stages, the closest predicted LAI values to ground-LAI were obtained using the GPR.

In maize (C2), GPR showed a LAI dynamic very similar to the one measured in situ, while LM and LogIF$_d$ overestimated the LAI, especially during SE. During the C2 FD stage, GPR and LogIF$_d$ showed a similar behavior, both underestimating the predicted LAI values, while LM exhibited a slight overestimation tendency. In alfalfa (C3), during the earliest vegetative (Vg) stage, LM showed the closest predicted LAI values to ground-LAI. During the Fl stage, GPR and LM showed similar performance.

## 4. Discussion

Considering spectral bands (e.g., near-infrared), indicators (e.g., NDVI), and final useful products (e.g., land-cover), there is a need to obtain valuable information for the development of LAI, as elaborated for the Copernicus indictors and products. Thus, assessment of the statistical relationship between ground-LAI and satellite remote sensing data for LAI prediction requires assessing the accuracy of LAI retrieval methods, according to ground-LAI estimation errors [42,57]. In this study, we measured ground-LAI for three crops, and showed a wide range of values, compared with those reported in the literature. Specifically, Revill et al. [58], using SunScan, exhibited a lower range of ground-LAI values (min, 0.5; max, 3.5) for winter wheat. However, our field data covered the entire cycle, from emergence to maturity, and agreed with Upreti et al. [26] (min, 1; max, 6.5).

Regarding maize, Facchi et al. [59], using LAI-2000, a ceptometer, and a Hemispherical camera in an experiment to compare (with destructive sampling measurements) the range of maize ground-LAI, the results were comparable to those we sampled in the field (min, 1; max, 5). Finally, for alfalfa, Verger et al. [60] measured a range of ground-LAI (min, 0.8; max, 6.5) that was lower with respect to our collected maxima (LAI > 8) using the LAI-2000 instrument. From this literature comparison, the only anomalous apparent difference of ground-LAI values was observed for alfalfa. However, the average value was coherent for different ESU in the same field for the 2018 sampling at flowering date (just before mowing) and, so, it was not in disagreement with the analysis. This difference, with respect to the literature, could arise from the uncertainty due to the optical instrument, the phenological stage, and the crop reflectance response [61,62].

Uncertainty of LAI estimations could arise from many factors, including crop type, farm management, and temporal variability of ground-LAI measurements [42,63].

In this work, we observed that, when using parametric approaches, SeLI proved to be the most suitable VI for LAI prediction under a mixed crop scenario. This result is in close agreement with previous findings highlighting that the combination of NIR and RE may provide more accurate LAI estimates for different crop types [18,34]. However, in the assessment of parametric methods, we also demonstrated that, at pixel level, the accuracy of LAI estimation was strongly affected not only by the VI selection but also (and not in a negligible manner) by the regression function as well as the parameterization data set [20,63]. The results of VIs also demonstrated the better LAI prediction ability of SeLI for wheat and maize, compared to alfalfa. The different LAI prediction ability of Vis, according to crop types, is in agreement with Herrmann et al. [64]. In this regard, Dong et al. [17], using RapidEye reflectance data and ground-LAI data of different crop types, demonstrated that VIs based on visible and RE regions are generally affected by chlorophyll, water content, and the structural properties of leaves and, therefore, the LAI predictability of a VI may vary markedly among crops and growing stages, according to canopy characteristics. Furthermore, when the crop-specific and mixed-crop parameterization were compared under both regression methods based on linear and logistic models (LM and LogIF$_d$), we demonstrated that the parameterization data set strongly influenced the LAI prediction and its accuracy. This result built on the study of Nguy-Robertson et al. [20] who, using hyperspectral data, found that RE-based VIs are little affected by crop type and, thus, may facilitate the prediction of LAI for different crops characterized by different canopy structure. This finding is very important for the future availability of

operational hyperspectral missions, such as the foreseen Copernicus CHIME and NASA SBG [31,65]. Thus, to obtain an accurate LAI prediction, the selection of suitable VIs is critical, as ideal VIs should be sensitive to the ground-LAI but insensitive to interference factors (e.g., the soil background, canopy structure, and chlorophyll content) [66]. Thus, despite the fact that SeLI yielded the most accurate results, it was also demonstrated that, due to the above-mentioned uncertainties, it might be not as accurate at pixel level.

Moreover, we showed how the three MLRAs (and, in particular, GPR) were able to exploit all information available from a multispectral data set, thus providing more accuracy in LAI prediction compared to VIs. Indeed, considering the relative weight of bands in the GPR algorithm, it was observed that, although the NIR and RE bands were the most relevant for LAI estimation, the other bands also contributed—varying according to the crop type and development stage—to LAI prediction. This finding is in agreement with previous outcomes. Delloy et al. [67] showed how the RE S2 bands (B5, B6, and B7) can improve winter wheat LAI estimation by using a hybrid approach (RTM + neural network). Verrelst et al. [13] compared different MLRAs, using simulated Sentinel-2 reflectance data over different crops types, and concluded that GPR was the most effective algorithm for LAI retrieval. However, despite the promising results of MLRAs, it was also observed that the performance was influenced by the training data set (i.e., MC vs. CS). This is in agreement with Mao et al. [25], who tested the influence of sample size on MLRA performances and highlighted how its influences the accuracy of the algorithms.

Regarding the temporal consistency of LAI estimation, our results showed that LAI retrieved by the parametric approach, at pixel level, was less suitable, with respect to the non-parametric approach. Indeed, the parametric method based on VIs showed a low accuracy, in terms of representing variability within the field, due to a saturation effect occurring especially at high vegetation density. In contrast, GPR allowed us to point out such variation, regardless of the crop development stage [58]. The low ability of parametric methods to account for the within variability of LAI was evidenced by the weak metrics of cross-validation carried out at pixel level. In particular, LM made it evident that parametric approaches may lead to the overestimation of LAI at early stages of crop development and underestimation at full canopy development. In this latter condition, VIs showed their limit in detecting LAI variation, due to the well-known issue of saturation [45]—a limit of VIs that was particularly exacerbated in the case of alfalfa, which reaches canopy closure, after resprouting, in less time compared to a winter cereal (wheat) and a row crop (maize). Conversely, GPR showed a high ability to detect LAI variation at the pixel level, regardless of the development stage, vegetation density, and crop type.

It is well-known that the canopy reflectance is affected by several biophysical and biochemical variables and, thus, the regions of the reflectance spectrum can be associated with different vegetation properties [68,69]. Therefore, exploitation of the full spectrum with non-parametric methods can improve the quality of LAI retrieval [13]. Indeed, our results demonstrated that the GPR outperformed the parametric methods; in addition, it was the most accurate MLRA for LAI prediction at both field and pixel level. The results of this study showed that the VI-based parametric method had a lower accuracy for LAI retrieval than MLRAs. These results suggest that GPR based on Sentinel-2 multispectral images is promising for crop monitoring, from a multi-crop mosaic scenario perspective. Further work will involve applying the MLRAs trained in this work to verify the model stability when applied to an independent data set; this analysis will allow for a full assessment of the robustness and exportability of the model developed.

The capacity of MLRAs to deal with full spectral information is a promising aspect that makes these approaches candidates for the investigation of new-generation hyperspectral data available from ASI-PRISMA, as well as those expected from the foreseen Copernicus CHIME and NASA SBG missions.

## 5. Conclusions

In the present study, the capability of different parametric and non-parametric methods for the retrieval of LAI for different arable crops using Sentinel-2 data was assessed. The accuracy and robustness of LAI estimates were compared, based on repeated in situ ground-LAI measurements throughout the crop growing season. Regarding the VI-based parametric regressions, the normalized index SeLI was evaluated as more suitable (i.e., than NDVI and NBR) for LAI retrieval at field level, providing good evaluation metrics by the cross-validation analysis for winter wheat and maize. However, VI-based parametric methods were shown: (i) to be unsuitable for LAI retrieval of alfalfa and mixed crop scenario; (ii) to have a very low accuracy for LAI retrieval at pixel level; and (iii) to have an accuracy of prediction the largely depends on VI selection, the fitting function, and the parameterization data set.

Among the non-parametric regression methods evaluated, the best-performing MLRA belonged to the kernel machine learning regression algorithms. Indeed, GPR was evaluated as the best-performing algorithm for LAI prediction, for the three arable crops evaluated. Using GPR, Sentinel-2 imagery can be used to map the spatial variability of the LAI of different arable crops, having prediction accuracy which is very high at the pixel level regardless of the crop type, growth stage, and the training data set. Moreover, GPR analysis of spectral bands in different phenological stages provided information on the relevance of relative bands in contributing to LAI prediction. However, further studies are required to fully assess the potential of GPR across different crops in different areas, as well as under contrasting agronomic conditions.

**Author Contributions:** Conceptualization, M.D.P., A.T. (Andrea Taramelli) and G.R.; methodology, M.D.P., A.T. (Andrea Taramelli), M.B., A.M., F.F. and G.R.; investigation, M.D.P., A.T. (Andrea Taramelli), M.B. and G.R.; data curation, M.D.P., A.M., F.F., I.V. and G.R.; writing—original draft preparation, M.D.P., A.M., F.F., I.V. and G.R.; writing—review and editing, M.D.P., A.T. (Andrea Taramelli), M.B., A.M., I.V., F.F., A.T. (Antonella Tornato), E.V. and G.R.; supervision, A.T. (Andrea Taramelli) and G.R. All authors have read and agreed to the published version of the manuscript.

**Funding:** The research was funded by the European Commission, ruled by Contract No. 4000125506/18/NL/IA "CHIME Mission Requirements Consolidation study", and the exploitation of results has been funded by the Italian Institute for Environmental Protection and Research (ISPRA) in the framework of agreement between ISPRA and Italian Space Agency (ASI) on "Air Quality" (Agreement number F82F17000000005) and by the Tuscany Region 2014-2020-Submeasure 16.5 PANACEA (CUP 787563).

**Institutional Review Board Statement:** Not applicable.

**Informed Consent Statement:** Not applicable.

**Data Availability Statement:** Data supporting the findings of this study are available on request from the corresponding author.

**Acknowledgments:** The authors wish to thank the staff from SSSA for trial management and ISPRA for their valuable support in the pre-processing stage of Sentinel-2 data.

**Conflicts of Interest:** The authors declare no conflict of interest.

# Appendix A

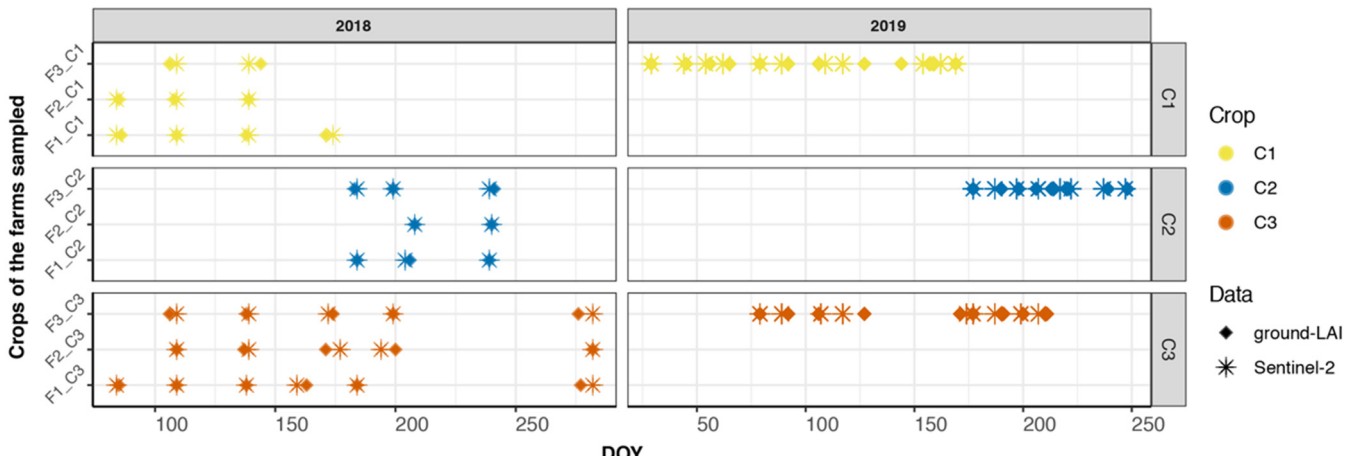

**Figure A1.** Sentinel-2 data (total, 37 images) acquired during the 2018 and 2019 crop seasons (C1, winter wheat; C2, maize; C3, alfalfa) and the ground-LAI acquisitions in the experimental sites F1, F2, and F3, around Pisa; DOY is day of the Year.

**Table A1.** Wavelength and spatial resolution of Sentinel-2 spectral bands.

| Spatial Resolution | Band | Name | Central WavelengthSentinel-A (nm) | Central WavelengthSentinel-B (nm) |
|---|---|---|---|---|
| Resolution 10 m: | B02 | Blue | 492.4 | 492.1 |
| | B03 | Green | 559.8 | 559.0 |
| | B04 | Red | 664.6 | 664.9 |
| | B08 | NIR | 832.8 | 832.9 |
| Resolution 20 m: | B05 | Red Edge 1 | 704.1 | 703.8 |
| | B06 | Red Edge 2 | 740.1 | 739.8 |
| | B07 | Red Edge 3 | 782.8 | 779.7 |
| | B8A | Vegetation Red Edge | 864.7 | 864.0 |
| | B11 | SWIR 1 | 1613.7 | 1610.4 |
| | B12 | SWIR 2 | 2202.4 | 2185.7 |
| Resolution 60 m: | B01 | Aerosols | 442.7 | 442.2 |
| | B09 | Water vapor | 945.1 | 943.2 |
| | B10 | Cirrus | 1373.5 | 1376.9 |

**Table A2.** Standard configuration of MLRAs using the ARTMO toolbox.

| Parameter | GPR | BOOST | BAGTREE |
|---|---|---|---|
| fit method | exact | | |
| basis function | constant | | |
| computational method | qr | | |
| kernel function | Squared Exponential Kernel | | |
| maximum number of trees | | 200 | 200 |

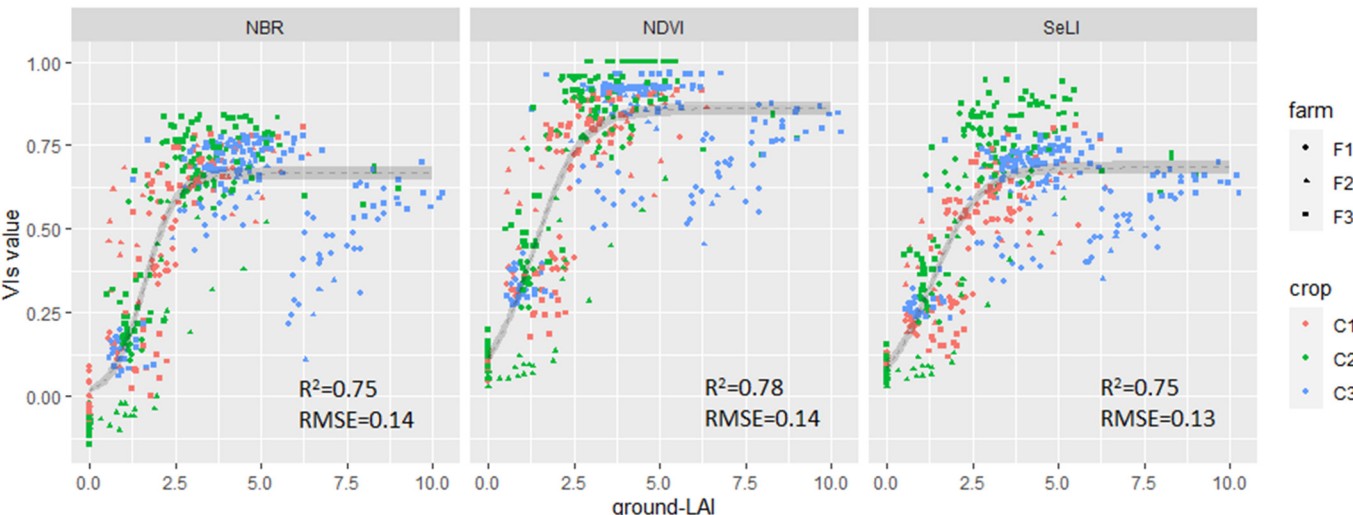

**Figure A2.** Non-linear regression between ground-LAI and VIs (NBR, NDVI, and SeLI), according to the three-parameter logistic function (L.3). Dashed line corresponds to the regression curve, while colored dots are the three crops: Yellow for winter wheat (C1), blue for maize (C2), and orange for alfalfa (C3). The shapes represent the farms (F1, F2 and F3).

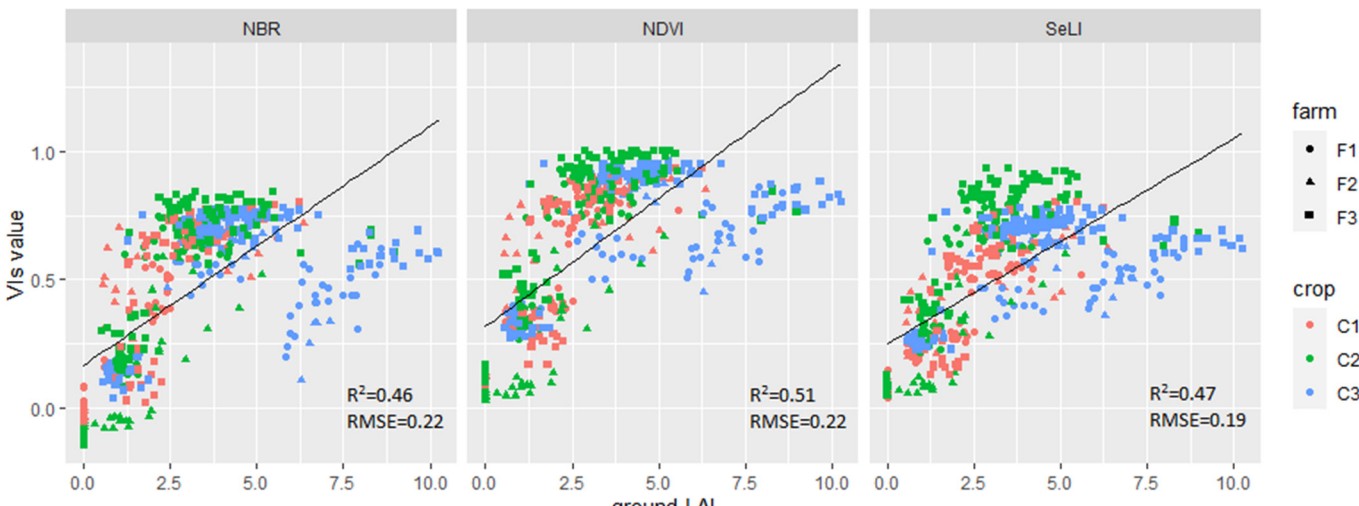

**Figure A3.** Linear regression between ground-LAI and VIs (NBR, NDVI, and SeLI), according to the Linear function (LM). Dashed line corresponds to the regression curve, while colored dots are the three crops: yellow for winter wheat (C1), blue for maize (C2), and orange for alfalfa (C3). The shapes represent the farms (F1, F2, and F3).

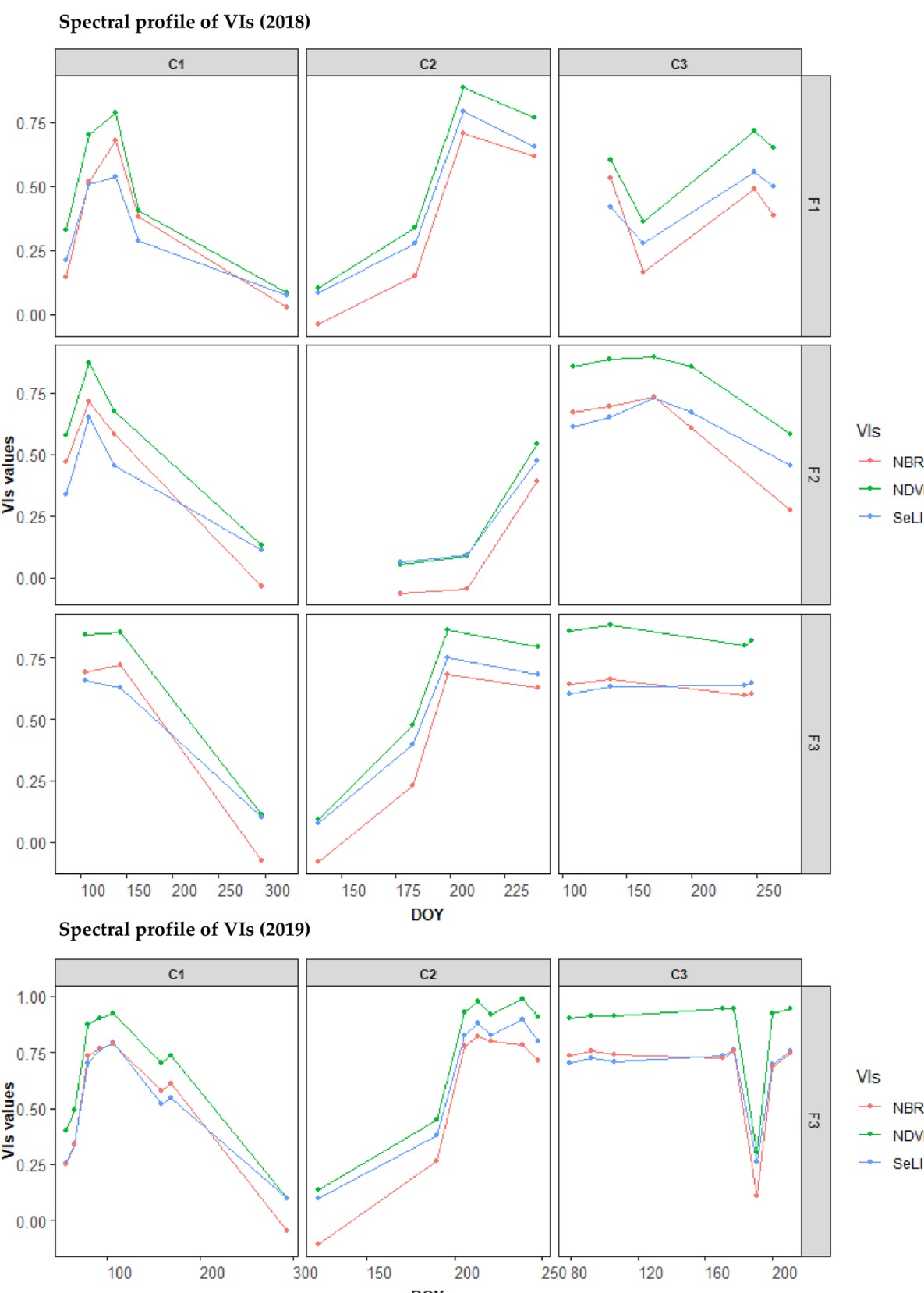

**Figure A4.** Multitemporal profile of VIs (NBR, NDVI, and SeLI) for the two-year ground-LAI sampling campaign.

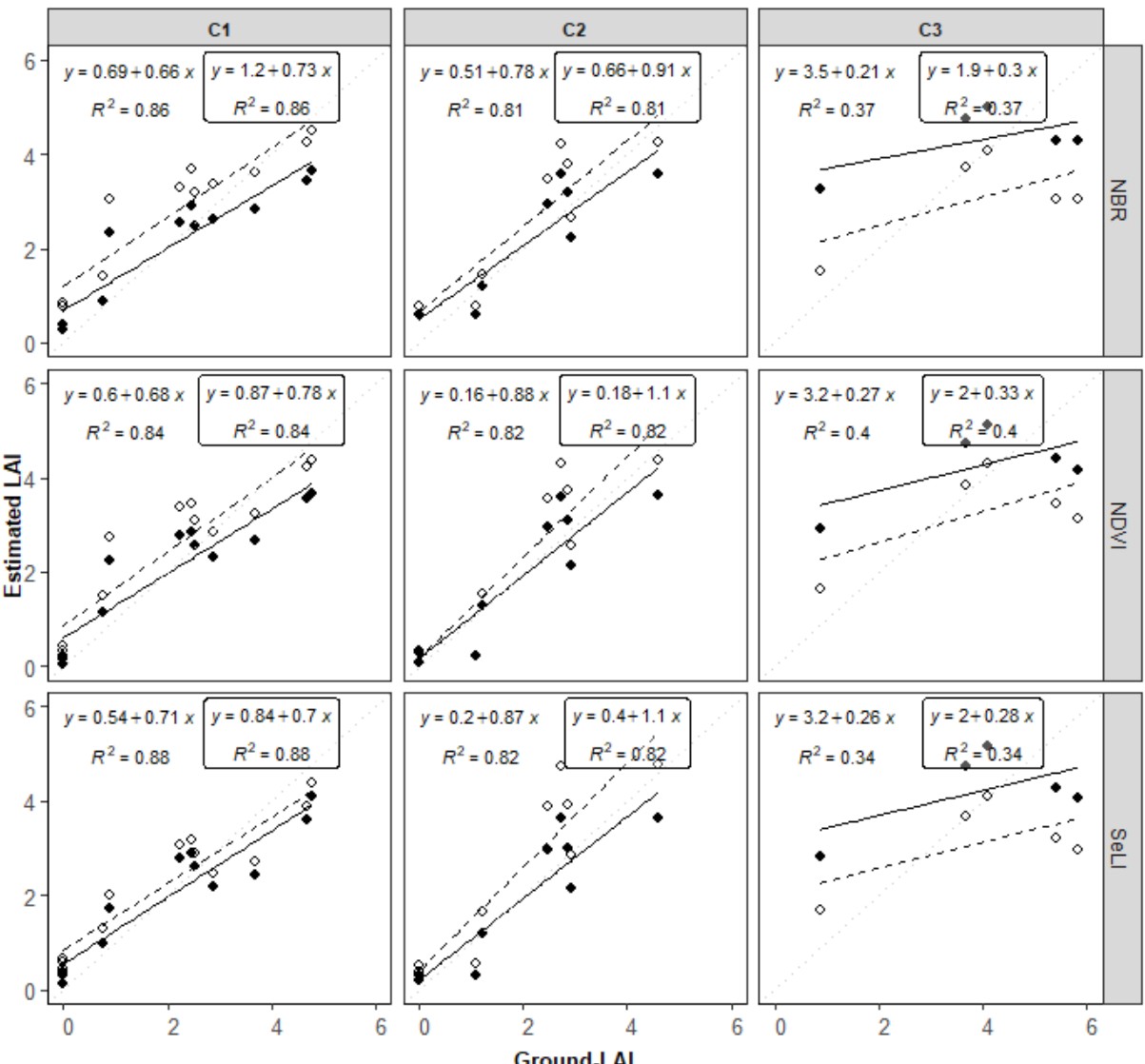

**Figure A5.** Results per crop (C1, C2, and C3) of linear regression analysis between measured ground-LAI and LAI predicted from NDVI, SeLI, and NBR by the mixed-crop (MC) LM (dashed line on black dots and equation in the box) and the crop-specific (CS) LM (continuous line on black dots).

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
