# Peer review of "Non-Parametric Statistical Approaches for Leaf Area Index Estimation from Sentinel-2 Data: A Multi-Crop Assessment"

_remotesensing, doi:10.3390/rs13142841_

Round 1
Reviewer 1 Report
This article presents an analysis between parametric and non-parametric LAI retrieval approaches from Sentinel-2 multispectral data for multiple crop types. These LAI retrieval methods, including simple vegetation indices and more complex machine learning algorithms, are evaluated based on ground measurements. The VI approaches were shown to perform well at the per-field scale, whereas the GPR machine learning algorithm performed better for estimating LAI at the sub-field scale (i.e., per-pixel basis) and appeared to be independent of crop growth stage. Although not mentioned in this manuscript, the GPR approach also allows for the propagation of uncertainty (i.e., uncertainty of the training data), thus maps of LAI from GPR can have corresponding maps of LAI uncertainty from the predictions. Such information is useful when applying probabilistic data assimilation approaches to crop models.
Overall, based on past research involving LAI retrieval approaches, the novelty of this particular article is not clear in the introduction section. I have suggested sentences throughout the manuscript should be checked for spelling and grammar. Care should also be taken with figure/table captions and the use of acronyms. I will not be recommending publication if this article at this stage.
Line 48-50: “…assist farmers to natural resource management” sentence unclear.
Line 51-52: Consider revising sentence. Also, please include reference for the LAI definition.
Line 54-55: Consider revising sentence, e.g. “… being a key state variable associated with processes including light interception and soil-crop water balance”.
Line 55: Remove “per se”.
Line 61: “..approaches for biophysical parameter retrieval …”?
Line 68-70: The key points from this sentence are not clear. Please consider revising sentence.
Line 74: Consider removing new paragraph and including on previous paragraph.
Line 84: “… in the last years …” “…past research has shown ..”
Line 91: “VIs-based” should be “VI-based” please check throughout manuscript.
Line 118: “To the scope …” consider removing.
Line 119: “…we built a database …”
Line 125: GPR was defined earlier.
Line 137: Please define MLRA.
Line 140: Not clear why “band sensitivity” is in quotation marks.
Line 190: Please provide a reference and definition of the BBCH scale. Not all readers would be familiar with this scale.
Line 206: “… of which 5 of bare soil …” I presume this was images taken during periods of bare soil (i.e. pre-emergence). Please be clear on this.
Line 227: Check formulation of the equations (Error!). Also check for other equations throughout the manuscript.
Line 231: Please define inflection point – not all readers will be familiar with the concept.
Line 233: “… according to CS and MC too”? Unclear.
Line 240: Please provide a brief description of the “the “drm” function”.
Line 287: “Sentinel-2 L2 data” should be “Sentinel-2 L2A data”?
Figure 4: Figure captions should be underneath the figure – please check this throughout the manuscript. Also, consider tidying up the text on the figures (i.e., overlapping text). It is not clear what is meant by the with/without box around the fit formula.
Figure 6: Nice idea about showing the Radar plots. It is, however, unclear about how the relative weight (%) was calculated – please provide this information in the methodology section.
Line 497: “…parametrization…” should be ”…parameterization…”? Check throughout the manuscript.
Author Response
Thank you for all the comments and suggestions, we really appreciated them. I attached all the comments.

Reviewer 2 Report
The overall objective of the paper is to evaluate the potential of non-parametric approaches for multi-temporal LAI retrieval from Sentinel-2 satellite data, in comparison with VIs-based parametric approaches.
The investigation highlights the shortcomings of vegetation indices used in parametric models and the benefits of using non-parametric models for multi-temporal and multi-crop LAI retrieval. This is important for the research community as VI are still commonly used for LAI retrievals from satellite data.
However the paper shows some clear shortcomings:
- The methodology section is sometimes a bit confusing and doesn’t justify clearly enough the choices taken.
- The result section sometimes lacks a bit of context in order to increase the understanding while reading. Also it is not clear to me why the band sensitivity is analysed after the non-parametric model - would it not make sense to do this prior?
Based on the general evaluation I recommend a minor revision.
I hope my remarks and comments attached to this review can help the authors to come to an improved manuscript.
Good luck!

Reviewer 3 Report
This study assessed parametric and non-parametric approaches to estimate LAI of wheat, corn and alfalfa from Sentinel-2 data. The experiments were well designed, the results were reasonable, the manuscript was well written. I have several comments before accecpting it for the publication.
1. The innovation of this study needs to be highlighted.
2. The "in-situ measurements" section need to be strengthened. First, is the SunScan derived LAI effective LAI (no clumping currection) or true LAI (with clumping correction)? Second, under what light conditions SunScan was used and what protocals were employed? Third, how many measurements were made within each ESU? Fourth, were ESUs same over the time for each field? Fifth, how were measurement-induced disturbances considered?
3. Scatter plots between VIs and LAI, and field-level or species-level time series of VIs for each crop type need to be provided at least in Supplementary materials. This can provide users basic and intuitive sense of crop growth conditions.
4. For machine learning approaches, the performance of training, testing and validation need to be provided at least in Supplementary materials. This can demonstrate whether the models were over-fitting.
5. How the hyperparameter were tuned need to be elaborated. The key parameter values need to be provided so that the comparison with other studies are possible.
6. I suggest incorporate multivariate linear regression into the study, because it also uses all bands in a simple manner. It lies between the simplest VI~LAI linear regression and the complex machine learning approaches, and thus can serve as a benchmark.
7. Why does GPR outperforms the other two machine learning approaches need to be discussed.
8. I'm pretty curious of the performance of different models on alfalfa when LAI > 7 ((e) in Figure Error! No text of specified style in document.)
9. Figure 7. Are black curves interpolated? If so, need to mark S2 observation dates.
10. The visualization of Figures and Tables can be improved. They do not look comfortable.
Reviewer 4 Report
This paper evaluated the machine learning regression algorithms (MLRAs) in the retrieval of leaf area index (LAI) by comparing it with conventional regression methods. Despite being a black/grey box model, machine learning certainly has great potentials in the interpretation of remote sensing products. This research contributes to the determination of high resolution LAIs. It is within the scope of Remote Sensing and meets the publication standards. In general, it is well written with coherent organization and solid arguments. Other than some apparent editorial errors in the numbers of equations and figures, I only have a few comments.
- Line 94-96, the authors pointed out that “a drawback of MLRAs methods is the instability when applied to datasets that deviated from the training dataset”. Indeed, this issues can be observed in Figure 7, LM actually outperforms GPR in the flowering (FI) for both crops C1 and C3. Are these ground measurements of LAIs in those cases considered as anomalies? If not, why the linear regression outperforms GPR?
- In Table 1, despite ‘a total of 558 representative ESUs were collected over the study are’, for each crop, the total ESUs from all three farms are less than 200. Among these ESUs, one third would be used as validation as k-fold cross validation was used. So basically only around 130 ESUs were used for training in MLRAs. Is this number sufficient? Was there any sensitivity analysis done regarding the size of training samples?
- The introduction of non-parametric methods (section 2.3.2) should be beefed up. Even though the implementation was done with software package, there is a lack of explanation of basic theories behind those methodologies.
- Figure 3 (I assume, as the figure number is unreadable), for the mean values of measured ground-LAI values, why the variation of C2_2018 in F3 is so large, was there any quality control for the ground measurement of LAIs?
Reviewer 5 Report
The topic of the manuscript is interest to the journal. It is generally clear. My comments that may be helpful for its revision include,
- Please provide more detail about the Grubbs test used in the database, and explanations on 232 samples were excluded.
- Line 176 – 179 how about the models when the bare soil samples were not considered.
- Line 211, LAI would change a lot within 10 day-gap. Thus, the selection did not make sense. Interpolating LAI observations may be a better choice.
- Table 2, why not use the B8A with a narrower wavelength range?
- Section 2.3.2.1 It is not clear how to get hyperparameter value for the k-fold cross-validation used in the model training.
- The quality of the three MLRAs would be largely affected by the number of samples when the MLRAs were trained for crop-specific subsets.
- Detail about validating LAI estimation may be required. It is unclear why very few LAI samples were used in the validation (Figure 4 - 5).
Round 2
Reviewer 1 Report
I appreciate the authors taking the time to address my concerns. I would now reccommend publicaiton of this article.
Author Response
Thank you again for the revisions.
Reviewer 3 Report
Most of my comments have been addressed and the manuscript has been improved. However, I'm still quite concerned for the ground measurements.
I have been using different types of instruments to measure different crop LAI for years. It is well known that line quantum sensor could give underestimates during clear days due to direct sunlight. Previous studies have shown that the underestimation of LAI could be up to 30% compared to LAI-2200 and DHP. How could you get LAIeff > 9 using SunScan in direct light?
Author Response
Comments and Suggestions for Authors
Most of my comments have been addressed and the manuscript has been improved. However, I'm still quite concerned for the ground measurements.
I have been using different types of instruments to measure different crop LAI for years. It is well known that line quantum sensor could give underestimates during clear days due to direct sunlight. Previous studies have shown that the underestimation of LAI could be up to 30% compared to LAI-2200 and DHP. How could you get LAIeff > 9 using SunScan in direct light?
Thank you again for this comment. Alfalfa ground-LAI > 8 refer to only one campaign immediately occurred before mowing when alfalfa was at flowering stage. In this case, the SunScan provided for the entire field coherent measurements of effLAI > 8 and for this reason we decided to keep the data in the dataset. However, we were aware that the measured effLAI were higher compared with the literature (Verger et al., 2009). In general, we had encountered several issues with alfalfa data interpretation, for both measurements and remote sensing (RS) estimations. Since RS estimations issues were widely discussed in the paper, we added a comment about the uncertainties of alfalfa effLAI measurements (Line 516).
Verger, A.; Martínez, B.; Camacho-De Coca, F.; García-Haro, F.J. Accuracy assessment of fraction of vegetation cover and leaf area index estimates from pragmatic methods in a cropland area. International Journal of Remote Sensing 2009, 30, 2685–2704, doi:10.1080/01431160802555804.
Reviewer 5 Report
Good to see the revision. However, the authors may not well address each comment I raised. In particular, I suggest the authors re-carefully read the publication about the SeLI index (Multi-Crop Green LAI Estimation with a New Simple Sentinel-2 LAI Index (SeLI), https://www.mdpi.com/1424-8220/19/4/904), and made a further revision. Thank you!
Author Response
Comments and Suggestions for Authors
Good to see the revision. However, the authors may not well address each comment I raised. In particular, I suggest the authors re-carefully read the publication about the SeLI index (Multi-Crop Green LAI Estimation with a New Simple Sentinel-2 LAI Index (SeLI), https://www.mdpi.com/1424-8220/19/4/904), and made a further revision. Thank you!
Thank you again for this comment. After re-reding my answer at this comment, I realized that I had replied in the wrong way. For SeLI we used the original formula of the authors (Pasqualotto et al., 2019). The B08 was used for the NDVI and NBR. In particular, we decided to use the B08 for NDVI calculation because the B08 and B04 have the same resolution (10m). Sorry again for the previous incorrect answer. Now we have corrected the Table 2.